# Integrative Analysis of *Oleosin* Genes Provides Insights into Lineage-Specific Family Evolution in Brassicales

**DOI:** 10.3390/plants13020280

**Published:** 2024-01-18

**Authors:** Zhi Zou, Li Zhang, Yongguo Zhao

**Affiliations:** 1National Key Laboratory for Tropical Crop Breeding, Hainan Key Laboratory for Biosafety Monitoring and Molecular Breeding in Off-Season Reproduction Regions, Institute of Tropical Biosciences and Biotechnology/Sanya Research Institute of Chinese Academy of Tropical Agricultural Sciences, Haikou 571101, China; zhangli0624@mail.scuec.edu.cn; 2Hubei Provincial Key Laboratory for Protection and Application of Special Plants in Wuling Area of China, College of Life Science, South-Central University for Nationalities, Wuhan 430074, China; 3College of Biology and Food Engineering, Guangdong University of Petrochemical Technology, Maoming 525011, China

**Keywords:** whole-genome duplication, gene expansion, evolutionary analysis, synteny analysis, orthogroup, divergence

## Abstract

Oleosins (OLEs) are a class of small but abundant structural proteins that play essential roles in the formation and stabilization of lipid droplets (LDs) in seeds of oil crops. Despite the proposal of five oleosin clades (i.e., U, SL, SH, T, and M) in angiosperms, their evolution in eudicots has not been well-established. In this study, we employed Brassicales, an economically important order of flowering plants possessing the lineage-specific T clade, as an example to address this issue. Three to 10 members were identified from 10 species representing eight plant families, which include Caricaceae, Moringaceae, Akaniaceae, Capparaceae, and Cleomaceae. Evolutionary and reciprocal best hit-based homologous analyses assigned 98 *oleosin* genes into six clades (i.e., U, SL, SH, M, N, and T) and nine orthogroups (i.e., U1, U2, SL, SH1, SH2, SH3, M, N, and T). The newly identified N clade represents an ancient group that has already appeared in the basal angiosperm *Amborella trichopoda*, which are constitutively expressed in the tree fruit crop *Carica papaya*, including pulp and seeds of the fruit. Moreover, similar to Clade N, the previously defined M clade is actually not Lauraceae-specific but an ancient and widely distributed group that diverged before the radiation of angiosperm. Compared with *A. trichopoda*, lineage-specific expansion of the family in Brassicales was largely contributed by recent whole-genome duplications (WGDs) as well as the ancient γ event shared by all core eudicots. In contrast to the flower-preferential expression of Clade T, transcript profiling revealed an apparent seed/embryo/endosperm-predominant expression pattern of most *oleosin* genes in *Arabidopsis thaliana* and *C. papaya*. Moreover, the structure and expression divergence of paralogous pairs was frequently observed, and a good example is the lineage-specific gain of an intron. These findings provide insights into lineage-specific family evolution in Brassicales, which facilitates further functional studies in nonmodel plants such as *C. papaya*.

## 1. Introduction

Oleosins are a class of highly abundant structural proteins of lipid droplets (LDs), which represent a major carbon reserve and are widely present in various plant organs such as seeds, pollen, flowers, fruits, and certain tubers [1,2,3,4,5]. Oleosins are typical for their small molecular weight (MW) of 14–30 kDa [5,6,7,8,9,10,11,12,13]. Nevertheless, all of them share a conserved central hydrophobic portion of approximately 72 residues, which could form a hairpin penetrating the surface phospholipid monolayer of an LD into the matrix. The hydrophobic hairpin is composed of two arms (each of about 30 residues) connected by a 12-residue loop with the pattern of PX_5_SPX_3_P, where X represents a nonpolar residue. By contrast, N- and C-terminal peptides, which lie on the phospholipid surface and may act as a receptor for metabolic enzymes or regulatory proteins, are amphipathic and usually variable [8,14]. Genome-wide surveys reveal that *oleosin* genes have already appeared in the single-celled algae, e.g., *Chlamydomonas reinhardtii*, and have diverged into at least six clades known as P (primitive), U (universal), SL (seed low), SH (seed high), T (tapetum), and M (mesocarp) during later evolution [4,8,15]. The most primitive Clade P was only found in green algae, mosses, and ferns, whereas Clade U, which is typical for the C-terminal AAPGA, is universally present in all land plants including *Selaginella moellendorffii*. Clade SL, which is present in seeds of both gymnosperms and angiosperms, was named after the low MW. This clade was proposed to first evolve from Clade U and later gave rise to Clades SH, M, and T. Clade SH, which is usually present in seeds of angiosperms, is typical for the high MW and C-terminal insertion relative to Clade SL. By contrast, Clades M and T were reported to be lineage-specific, which are confined to Lauraceae and Brassicaceae, respectively [4,8]. Comparative genomics analyses indicated that, for most clades, gene expansion was mainly contributed by whole-genome duplications (WGDs) especially those lineage-specific recent WGDs, e.g., the Brassicaceae-specific α WGD and the ρ WGD shared by cassava (*Manihot esculenta*) and rubber tree (*Hevea brasiliensis*) in Euphorbiaceae [5,6,12], in stark contrast to a key role of tandem duplication for Clade T in Brassicaceae [3,8,16].

Brassicaceae belongs to the order Brassicales, which includes 17 families, 398 genera, and 4450 species that have experienced multiple independent WGDs [17]. Thus far, genome-wide identification of *oleosin* family genes has been reported in 10 species within Brassicales. However, most of them (80%) belong to the Brassicaceae family [3,8,10]. Although it was established that Clade T is absent from papaya (*Carica papaya*, Caricaceae) and spider flower (*Tarenaya hassleriana*, Cleomaceae) [8], whether it is present or has been lost in other families within Brassicales is yet to be addressed. Recently available or updated genome assemblies for species in five Brassicales families beyond Brassicaceae, i.e., papaya [18], horseradish (*Moringa oleifera*, Moringaceae) [19], *Bretschneidera sinensis* (Akaniaceae) [20], caperbush (*Capparis spinosa*, Capparaceae) [21], *Cleome violacea* (Cleomaceae), acaya (*Gynandropsis gynandra*, Cleomaceae) [22], and spider flower [23], provide a good chance to uncover lineage-specific evolution of the *oleosin* gene family in this important plant order.

This study presents a comprehensive comparative analysis of the *oleosin* gene family in Brassicales. Significantly, our results showed that Clade M is actually not Lauraceae-specific but an ancient group that has already been present in the basal angiosperm *Amborella trichopoda* and is preserved in the early-diverging eudicot *Aquilegia coerulea* and all Brassicales species examined in this study. Moreover, a novel but ancient group named N was identified in most tested species, i.e., *A. trichopoda*, papaya, horseradish, *C. violacea*, acaya, and spider flower. In papaya, an economically and nutritionally important tree fruit crop widely cultivated in tropical and subtropical areas [18], this group was shown to be constitutively expressed, which includes pulp and seeds of the fruit. Herein, we report our findings.

## 2. Results

### 2.1. Identification of Oleosin Genes in A. trichopoda, Avocado, A. coerulea, and Representative Brassicales Species

To gain insight into lineage-specific family evolution in Brassicales, recently available chromosome (Chr)-level genome assemblies of *A. trichopoda* (a single living representative within the sister lineage Amborellales to all other flowering plants) [24], avocado (*Persea americana*, a Laurales member of an early-branching lineage of angiosperms that includes one M oleosin) [25], and *A. coerulea* (a Ranunculales member of the basal-most eudicot clade) [26] were first employed to identify *oleosin* family genes, resulting in five, three, and five members, respectively (Table 1). Five members identified in *A. trichopoda* and *A. coerulea* are consistent with what is found in previous assemblies [8], whereas only two avocado *oleosin* genes (i.e., *PaOLE2* and -*3*) have been reported by previous studies [4,27]. Moreover, an allele for *PaOLE2* that was discarded for further analyses in this study was also identified from tig00003364, and their coding sequences (CDS) were shown to exhibit 98.8% sequence identity, including only five single nucleotide polymorphisms (SNPs). Further mining genomes of representative Brassicales species resulted in six to 10 family members from papaya, horseradish, *B. sinensis*, caperbush, *C. violacea*, acaya, and spider flower (Table 1). Notably, compared with the previous study [8], one more member was identified in both papaya and spider flower, which were named *CpOLE6* and *ThOLE8*, respectively (Table 1).

Physiochemical parameters and conserved domains of deduced oleosin proteins are summarized in Table 1. In contrast to the great majority of oleosins featuring a single oleosin domain, MoOLE6 harbors two instead. Since the sequence was also found in two other genome assemblies [28,29], it is more likely to be a true gene that was resulted from tandem duplication. The sequence length of oleosins varies from 115 (CsOLE3) to 267 (MoOLE6) amino acids (AA) with an average of 151 AA, and correspondingly, their theoretical MW varies from 11.92 (CsOLE3) to 28.01 (MoOLE6) kDa with an average of 16.01 kDa. It is worth noting that CpOLE6, MoOLE6, CvOLE6, GgOLE7, and ThOLE8 possess unexpected low pI values of 4.43–6.56, in striking contrast to the alkaline characteristic of 9.23–11.00 for others. Except for BsOLE9, which exhibits an unusual GRAVY value of −0.144, the values for others are greater than 0, varying from 0.078 to 0.784 (Table 1). Nevertheless, all proteins possess relatively high aliphatic index (AI) values of 88.90–123.83 (Table 1) as well as similar Kyte–Doolittle hydrophobicity plots (except for MoOLE6) (Appendix A), which is in accordance with their amphipathic property.

### 2.2. Evolutionary Analysis and Definition of Orthogroups

To uncover their relationships, an unrooted evolutionary tree was first constructed using full-length protein sequences of five *AtrOLEs*, three *PaOLEs*, five *AcOLEs*, six *CpOLEs*, six *MoOLEs*, 10 *BsOLEs*, eight *CsOLEs*, six *CvOLEs*, seven *GgOLEs*, eight *ThOLEs*, eight *MeOLEs*, nine *PtOLEs*, and 17 *AtOLEs*. As shown in Figure 1A, they were clustered into six clades, five of which were previously defined as U, SL, SH, T, and M [4,8]. Whereas Clade T is restricted to Arabidopsis (*Arabidopsis thaliana*), Clade M, which was first described in the Lauraceae family [4,27], was unexpectedly found in all species examined in this study. The presence of Clade M in *A. trichopoda* (i.e., *AtrOLE2*) supports its early origin before the radiation of angiosperms. Moreover, a novel clade denoted N is not only present in papaya (i.e., *CpOLE6*), horseradish (i.e., *MoOLE6*), *C. violacea* (i.e., *CvOLE6*), acaya (i.e., *GgOLE7*), and spider flower (i.e., *ThOLE8*) but also in *A. trichopoda* (i.e., *AtrOLE5*), implying its early origin and lineage/species-species gene loss during later evolution. Structural features of Clade N relative to other CpOLEs are shown in Figure 1B. In contrast to AtrOLE5 possessing the conserved PX_5_SPX_3_P pattern, other members of Clade N exhibit PX_5_S/GPX_3_G/F variants. Moreover, an 18-residue insertion that is present in Clade SH was not detected in this clade as well as CpOLE4, MoOLE5, and CsOLE6, implying their divergence. Notably, AtrOLE4 possesses a 22-residue insertion instead (Figure 1B and Appendix A). Additionally, whereas the majority of U oleosins feature the C-terminal AAPGA, AcOLE1, and GgOLE1 harbor the AAPSA instead (Appendix A).

Furthermore, the BRH (best reciprocal hit) method was used to identify orthologs across different species. Except for T oleosins that were proven to be widely present in Brassicaceae plants [16], the criterion of at least one member present in more than one species examined in this study was used to define orthogroups (OGs). As shown in Figure 2 and Appendix A, a total of nine OGs were obtained, i.e., U1/-2, M, SL, SH1/-2/-3, N, and T, where five *AtrOLE* genes belong to U1, M, SL, SH1, and N, respectively, supporting early diversification of this family in angiosperms. During later evolution, linage-specific expansion and concentration were found. Notably, only two OGs (i.e., U1 and M) are preserved in avocado, whereas four OGs (i.e., U1, M, SL and SH1) are retained in *A. coerulea* (Figure 2).

### 2.3. Analysis of Exon–Intron Structure

To learn more about structure divergence, the exon–intron structures were analyzed on the basis of revised gene models. As shown in Table 1, a single intron was found in 27 out of 64 identified *oleosin* genes, occupying approximately 42.19%, smaller than 88.24% found in Arabidopsis (*At*-*T8* represents the sole member possessing two intron) (Appendix A). These intron-containing genes belong to Clades U, SL, SH, and N, which seems to be independent. Notably, no intron was found in Clade M as well as any member of *A. trichopoda*, avocado, and *A. coerulea*, whereas one intron is present in all SL members of papaya, horseradish, *B. sinensis*, and other Brassicales species. Moreover, in *C. violacea*, acaya, and spider flower, all U and SH members harbor an intron, whereas *GgOLE7* represents the unique N member with one intron (Table 1). Interestingly, the intron position appears to be conserved within clades but differs between different clades. Whereas Clade SL features one intron immediately after the sequence encoding the hydrophobic hairpin, the intron found in Clade N is located at the C-terminus of the hydrophobic hairpin; the intron found in Clade SH is located before the hydrophobic hairpin; and the intron found in Clade U is located at the C-terminus of the proline knot. These results imply an independent and lineage-specific gain of an intron (Figure 1 and Appendix A).

### 2.4. Gene Localization, Synteny Analysis, and Lineage-Specific Family Evolution in Brassicales

Gene localization revealed that identified *oleosin* genes are distributed across two-to-six chromosomes of *A. coerulea*, avocado, *B. sinensis*, *A. trichopoda*, papaya, caperbush, and acaya, and five-to-six scaffolds (Scfs) of horseradish, *C. violacea*, and spider flower, respectively (Figure 3). Further analysis of gene duplication events resulted in 54 duplicate pairs. Whereas most duplicate pairs were characterized as dispersed repeats, *CpOLE2*/-*3* and *BsOLE6*/-*7* were characterized as transposed and tandem repeats, respectively (Figure 3). Interestingly, despite the presence of five *oleosin* genes in *A. trichopoda*, intra-synteny analysis showed that none of them is located within syntenic blocks, which is similar to that observed in *A. coerulea*, papaya, and acaya. By contrast, one, one, one, two, four, four, and four WGD duplicate pairs were identified in avocado (i.e., *PaOLE1*/-*2*), horseradish (i.e., *MoOLE4*/-*5*), *C. violacea* (i.e., *CvOLE4*/-*5*), spider flower (i.e., *ThOLE4*/-*5* and *ThOLE6*/-*7*), *B. sinensis* (i.e., *BsOLE3*/-*4*, *BsOLE5*/-*6*, *BsOLE8*/-*9*, and *BsOLE8*/-*10*), caperbush (i.e., *CsOLE3*/-*4*, *CsOLE3*/-*5*, *CsOLE6*/-*7*, and *CsOLE6*/-*8*), and Arabidopsis (i.e., *At*-*Sm1*/-*2*, *At*-*S3*/-*5*, *At*-*S1*/-*4*, and *At*-*S2*/-*4*), respectively (Figure 3 and Figure 4).

Inter–synteny analyses were further conducted between *A. trichopoda*, avocado, *A. coerulea*, papaya, and Arabidopsis. As shown in Figure 4A, *AtrOLE* genes were shown to have three, two, and one syntelogs in avocado, *A. coerulea*, and papaya, respectively, but none in Arabidopsis; *AcOLE* genes also harbor one and three syntelogs in avocado and papaya, respectively, but none in Arabidopsis. These results reflect a long time of evolution, as well as two additional rounds of WGDs and massive chromosomal rearrangements that occurred in Arabidopsis after the split with papaya [30]. Nevertheless, three out of six *CpOLE* genes (i.e., *CpOLE1*, -*3*, and -*4*) still have eight syntelogs in Arabidopsis, i.e., one-to-two and one-to-three, reflecting their close relationship and lineage-specific WGDs. It is worth noting that, besides *At*-*S3* and *At*-*S5*, both *At*-*T1* and *At*-*T8* were also characterized as syntelogs of *CpOLE3*, which provides direct evidence for the origin of Clade T from Clade SL. Additionally, *PaOLE3*, a well-identified M member [4,27], still has syntelogs in *A. trichopoda* (i.e., *AtrOLE2*) and *A. coerulea* (i.e., *AcOLE2*) (Figure 4A), whereas *AcOLE2* still has syntelogs in poplar (*Populus trichocarpa*) (i.e., *PtOLE2a*/-*2b*) and cassava (i.e., *MeOLE2*) (Figure 4B).

In addition to *CpOLE1*, -*3*, and -*4*, *CpOLE2* and -*6* were also shown to have syntelogs in at least one species of horseradish, *B. sinensis*, caperbush, *C. violacea*, acaya, and spider flower (Figure 4C,D). Though no syntelog was identified for *CpOLE5* in all examined species, its orthologs *MoOLE5*, *BsOLE9*, *BsOLE10*, and *MeOLE5* are still located within syntenic blocks (Appendix A), implying a species-specific transposition of *CpOLE5*. Moreover, *MoOLE4*/-*5*, *BsOLE8*/-*9*/-*10*, and *MeOLE4b*/-*5* were also shown to be located within syntenic blocks, implying that two groups were derived from one WGD shared by these species, probably the γ event. Additionally, *CsOLE6*/-*7*/-*8*, *CvOLE4*/-*5*, *ThOLE6*/-*7*, and *At*-*S1*/-*2*/-*4* are also located within syntenic blocks. In fact, *CpOLE4* and -*5* exhibit a Ks value of 2.2048 (Appendix A), which is comparable to that of *MoOLE4*/-*5* (2.0410) and *CvOLE4*/-*5* (1.9437) (Table 2). However, this value is relatively higher than 1.5864 of *BsOLE8*/-*10* and 1.7862 of *MeOLE4b*/-*5*, implying a different evolutionary rate of γ WGD-derived repeats in these species. Similar cases were also observed for recent WGD repeats. Among four β WGD repeats identified in Brassicales species, *CsOLE3*/-*4* and *ThOLE6*/-*7* exhibit similar Ks values of 1.5677–1.6273, in contrast to high sequence divergence of *CsOLE7*/-*8* and *At*-*S1*/-*4*. As for three α WGD repeats identified in Arabidopsis, *At*-*Sm1*/-*2* and *At*-*S3*/-*5* exhibit similar Ks values of 1.3093–1.3683, which is relatively smaller than 1.5782 between *At*-*S1* and -*4*. By contrast, the Ks values of other recent WGD repeats identified in Brassicales species were relatively smaller, varying from 0.1962 to 0.5869, which is comparable to 0.1619–0.3696 of four p WGD repeats found in poplar and relatively smaller than 0.4175–0.7428 of three ρ WGD repeats identified in cassava (Table 2). In addition to *CpOLE4*/-*5*, three other dispersed repeats may also be derived from WGDs: *BsOLE1*/-*2* exhibit a Ks value of 0.1713, which is comparable to three α WGD repeats identified in *B. sinensis*, i.e., 0.1962–0.2810; *GgOLE1*/-*2* and *ThOLE1*/-*2* possess the Ks value of 0.6687 and 0.4058, respectively, which is comparable to that of the α WGD repeat *ThOLE4*/*5* (0.3409) but relatively smaller than the β WGD repeat *ThOLE6*/*7* (1.5677) (Table 2 and Appendix A). Notably, the Ka/Ks values of all repeats identified in this study were shown to be less than one, implying that they are subject to purifying selection.

### 2.5. Expression Divergence of Oleosin Genes

Global expression profiles of *AtOLE* genes were first examined from the Arabidopsis RNA-seq Database, which includes 28,164 libraries. As shown in Appendix A, most members of Clade T are preferential to be expressed in flowers, though *At*-*T8* is also expressed in embryos and seeds. By contrast, other members are predominantly expressed in seeds, embryos, and endosperms, as well as in silique. Notably, *At*-*Sm2* and *At*-*Sm3* were also shown to be expressed in pollen and flowers. Moreover, during embryo development, transcripts of most members (including *At*-*T8*) increase gradually, peaking at the stage of mature green. At the stage of 8-cell/16-cell, *At*-*S5*, *At*-*Sm2*, and *At*-*Sm1* represent the three most expressed isoforms, contributing 83.23% of total transcripts. Then, a sudden drop of total transcripts was observed at the globular stage, where *At*-*S5*, *At*-*Sm2*, and *At*-*Sm1* also contribute 75.98% of total transcripts. At stages from early heart to late torpedo, *At*-*S5* represents the most expressed isoform that contributes 44.69–62.58% of total transcripts. At the stage of bent cotyledon, *At*-*S1*, *At*-*S3*, and *At*-*S5* represent the three most expressed isoforms, contributing 76.94% of total transcripts. At the stage of mature green, *At*-*S3*, *At*-*S4*, and *At*-*S1* represent the three most expressed isoforms contributing 80.93% of total transcripts (Appendix A).

Then, papaya was used as an example of a fruit plant to study the expression evolution of *oleosin* genes. The RNA-seq data of various tissues, i.e., callus, shoot, hypocotyl, leaf, root, phloem sap, stamen, pollen, ovule, and pulp of mature fruit, were first investigated. As shown in Figure 5, their transcripts were detected in at least one of the tested tissues, though gene abundances are highly diverse. Total transcripts of the whole gene family were most abundant in shoot (100%), followed by callus (5.21–20.37%), and they were considerably low in other tissues (0.12–0.65%). In contrast to the constitutive expression of *CpOLE6*, *CpOLE1* was rarely expressed in sap and pulp. Whereas *CpOLE6* represents the unique isoform expressed in sap, three members were shown to be expressed in pulp, i.e., *CpOLE6*, -*3*, and -*5*. In the shoot and callus, *CpOLE3*, -*4*, and -*2* represent three dominant isoforms, which contribute 80.98–91.69% of total transcripts. On the contrary, *CpOLE4* was rarely expressed in other tissues; *CpOLE3* was rarely expressed in sap, stamen, pollen, and ovule; *CpOLE2* was rarely expressed in root, sap, and pulp; and *CpOLE5* was rarely expressed in root and sap. As expected, according to their expression patterns over various tissues, six *CpOLE* genes were grouped into three main clusters: Cluster I includes the two most expressed genes in shoot and callus, i.e., *CpOLE3* and -*4*; Cluster II includes two moderately expressed isoforms, i.e., *CpOLE2* and -*5*; and Cluster III includes *CpOLE6* and -*1*, which were constitutively expressed in most tissues (Figure 5A).

Since no transcriptome data are available for the seed tissue, qRT–PCR analysis was further conducted using seeds derived from mature fruits. As shown in Figure 5B, except for *CpOLE6* and -*1*, the expression levels of other *CpOLE* genes were significantly higher than the reference gene *CpEIEF*, varying from 2.61–36.81 folds, implying their divergence. Notably, *CpOLE3* and -*4* were shown to represent two dominant isoforms whose transcript levels were comparable (Figure 5B).

## 3. Discussion

The importance of oleosins in LD formation and stabilization has prompted active research in oil crops [31,32,33,34,35,36,37,38]. Nevertheless, despite the proposal of five oleosin clades (i.e., U, SL, SH, M, and T) in angiosperms [4,8], their evolution in eudicots has not been well-established. According to the comparison reported by Huang and Huang (2015), five *oleosin* genes present in *A. trichopoda* were assigned into two clades, i.e., U (1) and SL (4), though an M member was clearly identified [8]. Moreover, the distribution of the M clade, which was previously considered to be Lauraceae-specific [4], has not been well-studied.

In the present study, we used Brassicales, an economically important order of flowering plants that harbors the lineage-specific T clade [3,8,17], as an example to address evolution patterns of the *oleosin* gene family. In addition to 34 members reported in Arabidopsis, cassava, and poplar ([12], this study), a number of 64 *oleosin* family genes were identified from ten species representing eight plant families, i.e., Amborellaceae (*A. trichopoda*), Lauraceae (avocado), Ranunculaceae (*A. coerulea*), Caricaceae (papaya), Moringaceae (horseradish), Akaniaceae (*B. sinensis*), Capparaceae (caperbush), and Cleomaceae (*C. violacea*, acaya, and spider flower), while gene numbers of the family vary from three to ten. Interestingly, the family amounts are usually higher in species that experienced recent WGDs. According to comparative genomics analysis, after the split with *A. coerulea*, the last common ancestor of core eudicots underwent the γ whole-genome triplication (WGT) event at around 117 million years ago (MYA) [39]. Furthermore, Brassicaceae species, represented by Arabidopsis, experienced two more WGDs named At-β (60–65 MYA) and At-α (~35 MYA) [30,40], where the At-β WGD was shown to be shared by caperbush, *C. violacea*, acaya, and spider flower [17,21,22,23]. In the Capparaceae lineage, caperbush further experienced one independent WGD known as Cs-α at 18.6 MYA [21]. In the Cleomaceae lineage, after the split with *C. violacea*, the last common ancestor of acaya and spider flower first experienced one independent WGD known as Gg-α (~22 MYA), which was followed by an addition of a third genome (Th-α, ~18.4 MYA) to spider flower but not acaya [41]. After the split with papaya, *B. sinensis* in the Akaniaceae lineage was also shown to experience one independent WGD known as Bs-α [20]. Correspondingly, compared with five members present in both *A. trichopoda* and *A. coerulea*, one more was identified in papaya, horseradish, and *C. violacea*. By contrast, more than seven members were identified in *B. sinensis*, acaya, and spider flower, which are comparative to eight and nine reported in cassava and poplar, respectively [12].

According to evolutionary analysis, 98 *oleosin* genes were grouped into six clades, one more than that described before [4,8,12]. Interestingly, this novel and so-called N clade are present in *A. trichopoda* and most Brassicales species examined in this study, implying its early origin and lineage-specific gene loss. Besides Clade N, four other *AtrOLE* genes were assigned into four clades, i.e., U, SL, SH, and M, instead of only two as proposed by Huang and Huang (2015) [8]. The updated classification is not only supported by evolutionary analysis but also by BRH-based orthologous and synteny analyses. Whereas Clades SL, M, N, and T contain a single OG, U and SH have evolved to form two and three, respectively, a high member of which are still located within syntenic blocks. As for Clade M, *PaOLE3*, *AtrOLE2*, *AcOLE2*, *MeOLE2*, *PtOLE2a*, and *PtOLE2b* were shown to be located within syntenic blocks, whereas *CpOLE2*, *MoOLE3*, *BsOLE3*, *BsOLE4*, *CsOLE2*, *CvOLE2*, *GgOLE3*, *ThOLE3*, and *At*-*Sm3* were also characterized as syntelogs, implying a highly conserved evolution of this clade, which argues Lauraceae-specific distribution proposed by Huang and Huang (2016) [4]. Moreover, this clade has expanded in *B. sinensis* and poplar via recent WGDs, which were shown to be Akaniaceae and Salicaceae-specific, respectively [20,42]. As for Clade N, despite a frequent loss in species examined in this study, *CvOLE6*, *GgOLE7*, and *ThOLE8* are still located within syntenic blocks, implying possible functions in specific biological processes that are yet to be studied. As for Clade U, which is typical for the C-terminal AAPGA [8,12], gene expansion was observed in avocado, cassava, *B. sinensis*, acaya, spider flower, and Arabidopsis, which were contributed by WGDs and dispersed duplication. Among them, though *BsOLE1* and -*2* are no longer located within syntenic blocks, both of them were characterized as syntelogs of *CsOLE1*, which is consistent with their comparable Ks value to three Bs-α WGD repeats identified in this species, i.e., *BsOLE3*/*4*, *BsOLE5*/*6*, and *BsOLE9*/*10*, implying their WGD-derivation and chromosome rearrangement of the *BsOLE2*-encoding region. Similar cases were also observed for *GgOLE1*/-*2* and *ThOLE1*/-*2*, where *GgOLE1*, *GgOLE2*, and *ThOLE1* were characterized as syntelogs of *CsOLE1*, *At*-*Sm1*, and *At*-*Sm2*, though *ThOLE2* is no longer located within syntenic blocks. As for Clade SL, gene expansion was observed in *A. coerulea*, *B. sinensis*, caperbush, spider flower, Arabidopsis, cassava, and poplar, which were contributed by WGDs, as well as tandem and dispersed duplication. Notably, *BsOLE6* and -*7* represent the unique pair of tandem repeats beyond Clade T. Compared with other clades, Clade SH has extensively expanded in core eudicots, forming three OGs as identified in this study. Among them, SH1 and SH2 are more likely to arise from the γ event [39], and *MoOLE4*, *MoOLE*-*5*, *BsOLE8*, *BsOLE9*, *BsOLE10*, *MeOLE4a*, and *MeOLE5* are still located within syntenic blocks with similar Ks values, whereas SH3 appears to be generated by the At-β event [30]. Moreover, SH1 has further expanded in caperbush, Arabidopsis, cassava, and poplar via lineage-specific recent WGDs, i.e., Cs-α, At-α, ρ, and p, respectively [20,30,42,43]. It is worth noting that, despite the wide presence of Clade T in Brassicaceae plants [8,16], no ortholog was identified in any other Brassicales species examined in this study, implying its appearance sometime after the Brassicaceae–Cleomaceae divergence. Nevertheless, *At*-*T1* and *At*-*T8* were characterized as syntelogs of *CpOLE3*, *ThOLE4*, *ThOLE5*, *GgOLE4*, *CsOLE3*, *CsOLE4*, and *CsOLE5*, implying that Clade T was indeed derived from Clade SL.

In addition to species-specific retention of repeats after WGDs, structural divergence was also shown to play a role in the evolution of the *oleosin* family. In contrast to no intron that is present in *oleosin* genes of *A. trichopoda*, avocado, and *A. coerulea*, Clade SL has gained one intron immediately after the sequence encoding the hydrophobic hairpin stretch in all Brassicales species examined in this study, which is similar to that reported in Salicaceae and Euphorbiaceae [12]. Interestingly, the intron position found in Clade SL is different from that observed in several members of Clades U, SL, SH, and N, implying an independent gain of an intron. Since all SH members in *C. violacea*, acaya, spider flower, and Arabidopsis feature the intron that is located before the hydrophobic hairpin, its gain may occur sometime after the split with Capparaceae but before Brassicaceae–Cleomaceae divergence. The absence of the Cleomaceae U intron in Arabidopsis, which is located at the C-terminus of the proline knot, implies that its gain occurred sometime after the split with Brassicaceae. By contrast, the intron found in *GgOLE7*, which is located at the C-terminus of the hydrophobic hairpin, may be *Gynandropsis*-specific, since it is absent from its orthologs *CvOLE6* and *ThOLE8*.

Expression divergence also plays an important role in the evolution of *oleosin* family genes in Brassicales. Among six *oleosin* genes identified in papaya, *CpOLE6* in Clade N and *CpOLE1* in Clade U have evolved to be constitutively expressed, whereas *CpOLE3* in Clade SL and *CpOLE4* in Clade SH have evolved into two dominant isoforms in seeds, calluses, and shoots, though *CpOLE4* is more likely to be a WGD (γ) repeat of *CpOLE5*, another SH member. The constitutive expression of U *oleosin* genes has been widely reported in other species, e.g., castor bean (*Ricinus communis*), physic nut (*Jatropha curcas*), cassava, rubber tree, safflower (*Carthamus tinctorius*), rapeseed (*Brassica napus*), and tigernut (*Cyperus esculentus*) [5,9,10,11,12,13]. Nevertheless, to our surprise, *CpOLE1* was shown to be rarely expressed in both sap and pulp, which is different from *CpOLE6*. Compared with *CpOLE4*, transcript levels of *CpOLE5* were shown to be considerably lower in seeds, calluses, and shoots. By contrast, it was also moderately expressed in pollen, stamens, and ovules, as well as pulp. Notably, though Clade M was previously reported to be mesocarp-abundant [8,42], the expression of *CpOLE2* was rarely detected in pulp or roots and sap. Interestingly, the transcript level of *CpOLE2* is usually higher than that of *CpOLE5*, *CpOLE6*, and *CpOLE1* in most tissues. By contrast, its ortholog in Arabidopsis (*At*-*Sm3*) is always less expressed than most members beyond Clade T. Moreover, among several repeat pairs identified in Arabidopsis, i.e., *At*-*Sm1*/-*Sm2*, *At*-*S3*/-*S5*, *At*-*S1*/-*S2*/-*S4*, and *At*-*T1*/-*T2*/-*T3*/-*T4*/-*T5*/-*T6*/-*T7*/-*T8*/-*T9*, *At*-*Sm1*, *At*-*S5*, *At*-*S4*, and *At*-*T5* have evolved into dominant isoforms, respectively. In Brassicales, the lineage-specific expansion and tissue-specific expression of *oleosin* genes reflect their roles in the oil accumulation of seeds and anther [3,34]. In seeds, the accumulation of oleosins is usually negatively correlated with LD size but positively associated with oil content, which could not only affect seed germination but also the freezing tolerance of seeds [34,35]. Moreover, Brassicaceae-specific T oleosins are acquired for tapetosome formation, which confer additive benefits of pollen vigor [44].

## 4. Materials and Methods

### 4.1. Sequence Retrieval and Identification of Oleosin Family Genes

*Oleosin* genes reported in Arabidopsis (Brassicaceae, Brassicales), poplar (Salicaceae, Malpighiales), and cassava (Euphorbiaceae, Malpighiales) were updated according to references [6,12], and detailed information is shown in Appendix A. Genomic sequences of *A. trichopoda* (v2.1; Amborellaceae, Amborellales), avocado (Gwen v1; Lauraceae, Laurales), *A. coerulea* (v3.1; Ranunculaceae, Ranunculales), papaya (Sunset v1; Caricaceae, Brassicales), *B. sinensis* (v1; Akaniaceae, Brassicales), horseradish (v1; Moringaceae, Brassicales), caperbush (v1; Capparaceae, Brassicales), *C. violacea* (v2.1; Cleomaceae, Brassicales), acaya (v1; Cleomaceae, Brassicales), and spider flower (v1; Cleomaceae, Brassicales) were downloaded from public databases, i.e., Phytozome (v13, https://phytozome.jgi.doe.gov/pz/portal.html, accessed on 31 October 2023), NGDC (http://bigd.big.ac.cn/gsa, accessed on 31 October 2023), and NCBI (https://www.ncbi.nlm.nih.gov/, accessed on 31 October 2023). To identify oleosin homologs, the oleosin domain profile (PF01277) was used for HMMER (v3.3, http://hmmer.janelia.org/, accessed on 31 October 2023) searches as described before [45,46]. All predicted gene models were manually curated with available mRNAs, including nucleotides, Sanger-expressed sequence tags (ESTs), and RNA sequencing (RNA-seq) reads that were accessed from NCBI (accessed on 31 November 2023). Presence of the conserved oleosin domain in deduced peptides was confirmed using MOTIF Search (https://www.genome.jp/tools/motif/, accessed on 31 October 2023), whereas protein properties were calculated using ProtParam (http://web.expasy.org/protparam/, accessed on 31 October 2023). Additionally, pseudogenes and/or homologous fragments present in related genomes were also identified with CDS sequences of obtained *oleosin* genes as previously described [12].

### 4.2. Sequence Alignment, Evolutionary Analysis, and Definition of Orthogroups

Multiple sequence alignment was conducted using MUSCLE [47], which was subject to evolutionary tree construction using MEGA 6.0 [48] with the maximum likelihood method, Jones–Taylor–Thornton (JTT) model, uniform rates, complete deletion of gaps, nearest-neighbor interchange (NNI), and bootstrap of 1000 replicates. Orthologs between species were identified using the BRH (best reciprocal hit) method, and OGs across different species were defined as described before [49,50], which were assigned only when they were present in at least two species tested.

### 4.3. Gene Localization, Synteny Analysis, and Calculation of Evolutionary Rate

Gene locations on chromosomes and/or scaffolds were inferred from the revised genome annotation and displayed using TBtools [51]. For synteny analysis, duplicate pairs between or within species were identified using the all-to-all BLASTp [52] method with *E*-value cutoff of 1 × 10^−10^, and gene colinearity was inferred using MCScanX [53] with the cutoff of five BLAST hits. Duplication modes such as tandem, proximal, transposed, dispersed, and WGD were identified using the DupGen_finder pipeline as previously described [54], and Ks (synonymous substitution rate) and Ka (nonsynonymous substitution rate) of duplicate pairs were calculated using codeml [55].

### 4.4. Gene Expression Analysis

Expression profile data of *AtOLE* genes were accessed from Arabidopsis RNA-seq Database (https://plantrnadb.com/athrdb/, accessed on 31 October 2023) and Arabidopsis Embryo eFP Browser (https://bar.utoronto.ca/efp/cgi-bin/efpWeb.cgi, accessed on 31 October 2023), whereas global expression profiles of *CpOLE* genes were analyzed using transcriptome datasets as shown in Appendix A. Raw sequence reads in the FASTQ format were obtained using fastq-dump, and quality control was performed using Trimmomatic [56]. Read mapping was conducted using HISAT2 [57], and the FPKM (fragments per kilobase of exon per million fragments mapped) method was used to determinate relative transcript levels.

To uncover the relative expression levels of *CpOLE* genes in the seed tissue, mature seeds were collected from the yellow fruits of Zhongbai cultivar as described before [58]. Total RNA extraction, synthesis of the first-strand cDNA, and qRT–PCR analysis were conducted as previously described [59], where *CpEIEF* was used as the reference gene. Primers used in this study are shown in Appendix A. Relative gene expression levels were estimated with the 2^−ΔΔCt^ method, and statistical analysis was performed using SPSS Statistics 20 as described before [60].

## 5. Conclusions

In this study, a focus on a comparative analysis of the *oleosin* gene family in Brassicales was conducted, which includes 13 species representing 10 plant families. Ninety-eight *oleosin* genes were assigned into six clades (i.e., U, SL, SH, M, N, and T) and nine OGs (i.e., U1, U2, SL, SH1, SH2, SH3, M, N, and T). The newly identified Clade N represents an ancient group that diverged before the radiation of angiosperm. Interestingly, this group was constitutively expressed in papaya, which includes the fruit and sap. Moreover, the previously defined Clade M is not Lauraceae-specific but an ancient and widely distributed group that has already appeared in the basal angiosperm *A. trichopoda*. Compared with *A. trichopoda*, the family expansion in Brassicales was largely contributed by lineage-specific recent WGDs but also the ancient γ event shared by all core eudicots. The expression of Clade T was shown to be flower-preferential, whereas other members exhibit an apparent seed/embryo/endosperm-predominant expression pattern. The structure and expression divergence of paralogous pairs was frequently observed, and a good example is a lineage-specific gain of an intron. These findings provide insights into lineage-specific family evolution in Brassicales, which facilitates further functional studies in papaya and other nonmodel species.

## Figures and Tables

**Figure 1 plants-13-00280-f001:**
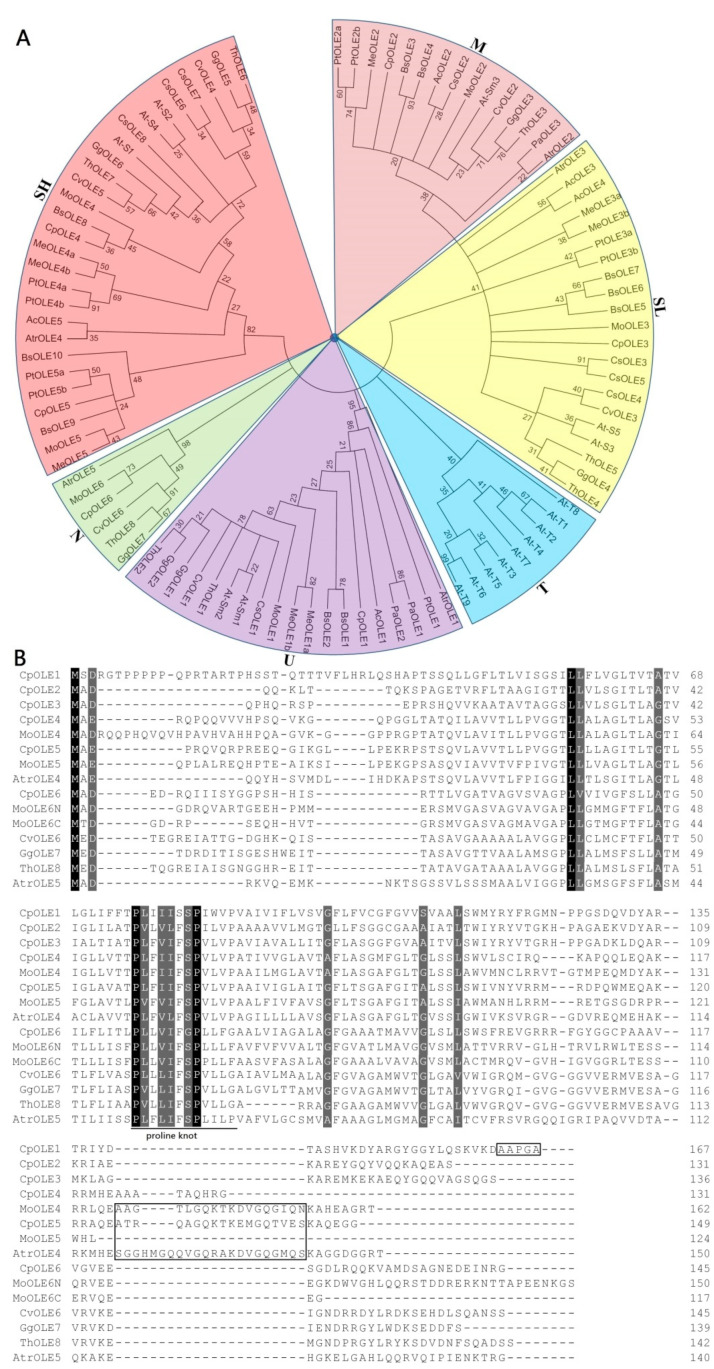
Multiple sequence alignment and evolutionary analysis of oleosins. (**A**) Evolutionary analysis of oleosins. Shown is an unrooted evolutionary tree resulting from full-length oleosins with MEGA6 (maximum likelihood method and bootstrap of 1000 replicates), where the distance scale denotes the number of AA substitutions per site and the name of each clade is indicated next to the corresponding clade. (**B**) Sequence alignment and structural features of N oleosins together with AtrOLE4, MoOLE4, MoOLE5, and other CpOLEs. MoOLE6N and MoOLE6C represent N- and C termini of the MoOLE6 protein, whereas sequence alignment and display were conducted using MUSCLE and Boxshade, respectively. Identical and similar residues are highlighted in black and dark grey, respectively. The conserved 12-residue proline knot is underlined, whereas the C-terminal AAPGA of Clade U and the putative C-terminal insertion of Clade SH are boxed. (Ac: *A. coerulea*; At: *A. thaliana*; Atr: *A. trichopoda*; Bs: *B. sinensis*; Cp: *C. papaya*; Cs: *C. spinosa*; Cv: *C. violacea*; Gg: *G. gynandra*; Me: *M. esculenta*; Mo: *M. oleifera*; M: mesocarp; N: novel; OLE: oleosin; Pa: *P. americana*; *P. trichocarpa*; SH: seed high-molecular-weight; SL: seed low-molecular-weight; Th: *T. hassleriana*; T: tapetum; U: universal).

**Figure 2 plants-13-00280-f002:**
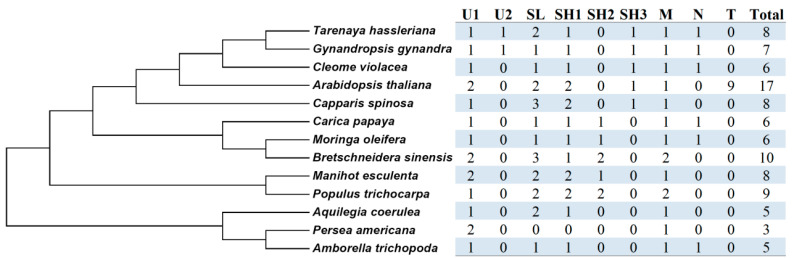
Species-specific distribution of nine oleosin orthogroups identified in this study. Taxonomy relationships of tested species follow that of NCBI Taxonomy (M: mesocarp; N: novel; SH: seed high-molecular-weight; SL: seed low-molecular-weight; T: tapetum; U: universal).

**Figure 3 plants-13-00280-f003:**
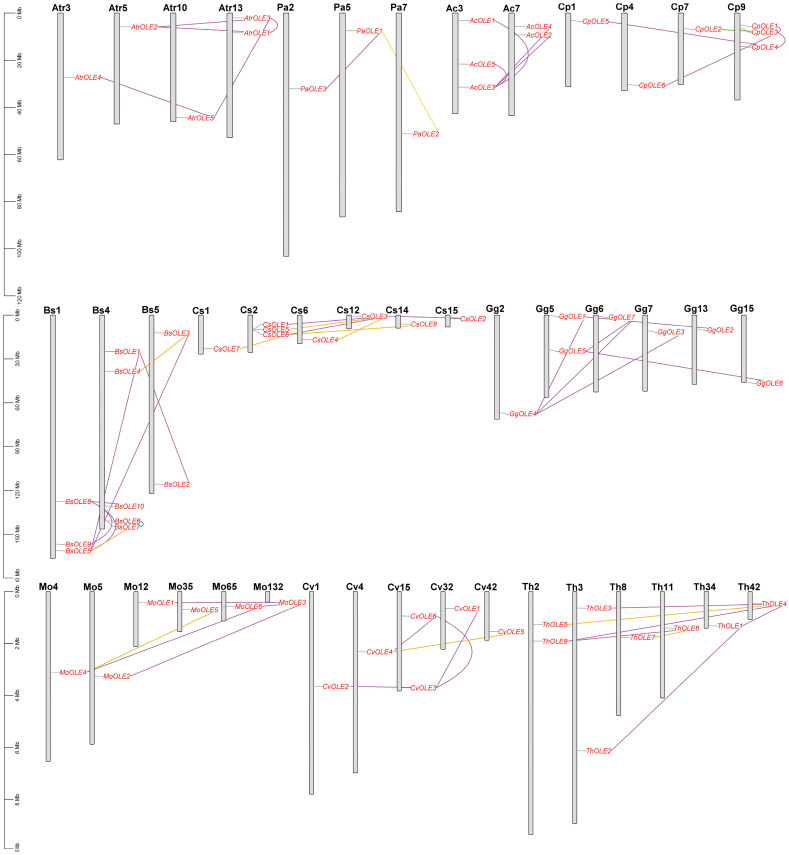
Chromosomal locations and duplication events of *oleosin* genes. Serial numbers are indicated at the top of each chromosome/scaffold, and the scale is in Mb. Duplicate pairs identified in this study are connected using lines in different colors, i.e., tandem (blue), transposed (green), dispersed (purple), and WGD (gold). (Ac: *A. coerulea*; Atr: *A. trichopoda*; Bs: *B. sinensis*; Chr: chromosome; Cp: *C. papaya*; Cs: *C. spinosa*; Cv: *C. violacea*; Gg: *G. gynandra*; Mo: *M. oleifera*; OLE: oleosin; Pa: *P. americana*; Scf: scaffold; Th: *T. hassleriana*).

**Figure 4 plants-13-00280-f004:**
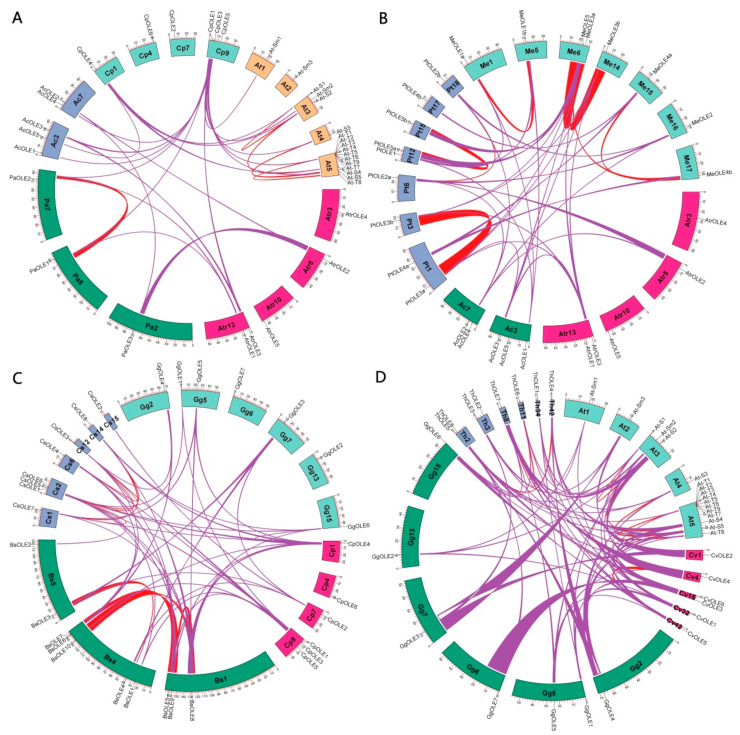
Synteny analyses within and between C. papaya and other species. (**A**) *C. papaya*, *A. thaliana*, *A. coerulea*, *P. americana*, and *A. trichopoda*. (**B**) *P. trichocarpa*, *M. esculenta*, *A. coerulea*, and *A. trichopoda*. (**C**) *C. papaya*, *B. sinensis*, *C. spinosa*; and *G. gynandra*. (**D**) *C. violacea*, *G. gynandra*, *T. hassleriana*, and *A. thaliana*. Syntenic blocks were inferred using MCScanX (E-value ≤ 1 × 10^−10^; BLAST hits ≥ 5). Oleosin-encoding chromosomes/scaffolds are shown, and only syntenic blocks that contain oleosin genes are marked in red (intra) and purple (inter), respectively. (Ac: *A. coerulea*; At: *A. thaliana*; Atr: *A. trichopoda*; Bs: *B. sinensis*; Chr: chromosome; Cp: *C. papaya*; Cs: *C. spinosa*; Cv: *C. violacea*; Gg: *G. gynandra*; Me: *M. esculenta*; Mo: *M. oleifera*; OLE: oleosin; Pa: *P. americana*; Pt: *P. trichocarpa*; Scf: scaffold; Th: *T. hassleriana*).

**Figure 5 plants-13-00280-f005:**
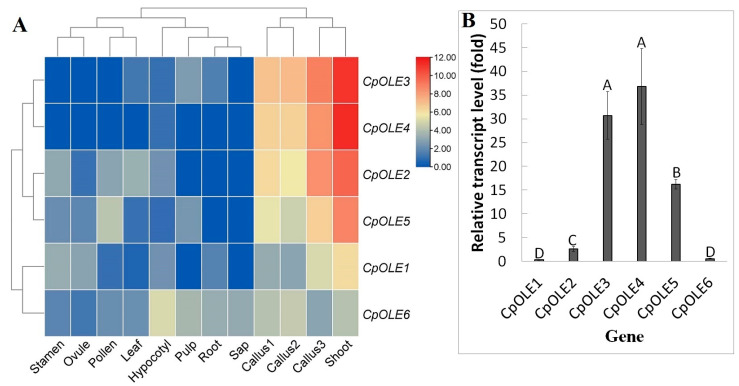
Expression profiles of *CpOLE* genes. (**A**) Tissue-specific expression profiles of *CpOLE* genes. Color scale represents FPKM normalized log_2_ transformed counts where blue indicates low expression and red indicates high expression. (**B**) *CpOLE* transcript abundance relative to the reference gene *CpEIEF*. Bars indicate SD (N ≥ 3) and uppercase letters indicate difference significance tested following Duncan’s one-way multiple-range post hoc ANOVA (*p* < 0.01). (Cp: *C. papaya*; FPKM: fragments per kilobase of exon per million fragments mapped; OLE: oleosin).

**Table 1 plants-13-00280-t001:** *Oleosin* genes identified in *A. trichopoda*, *P. americana*, *A. coerulea*, and representative Brassicales species. (AA: amino acid; Ac: *A. coerulea*; AI: aliphatic index; Atr: *A. trichopoda*; Bs: *B. sinensis*; Chr: chromosome; Cp: *C. papaya*; Cs: *C. spinosa*; Cv: *C. violacea*; Gg: *G. gynandra*; GRAVY: grand average of hydropathicity; II: instability index; kDa: kilodalton; Mo: *M. oleifera*; MW: molecular weight; OLE: oleosin; Pa: *P. americana*; pI: isoelectric point; Scf: scaffold; Th: *T. hassleriana*).

Gene Name	Locus	Position	Intron No.	AA	MW (kDa)	pI	GRAVY	AI	Duplicate	Mode	Oleosin Location	Clade
*AtrOLE1*	AmTrH2.13G041800	Chr13:8312320..8313509(−)	0	168	17.80	10.31	0.263	105.71	-	-	42..154	U
*AtrOLE2*	AmTrH2.05G030700	Chr5:5790540..5791413(+)	0	147	15.51	9.36	0.349	105.58	*AtrOLE1*	Dispersed	22..134	M
*AtrOLE3*	AmTrH2.13G011500	Chr13:2034840..2035253(−)	0	137	14.07	9.75	0.411	103.43	*AtrOLE1*	Dispersed	22..134	SL
*AtrOLE4*	AmTrH2.03G086400	Chr3:27204354..27205476(+)	0	150	15.56	9.36	0.365	104.60	*AtrOLE5*	Dispersed	21..136	SH
*AtrOLE5*	AmTrH2.10G130400	Chr10:44382338..44383409(−)	0	140	14.75	9.94	0.531	103.14	*AtrOLE3*	Dispersed	17..131	N
*PaOLE1*	g26506	Chr5:7463879..7464397(−)	0	172	17.99	10.01	0.294	100.41	-	-	47..157	U
*PaOLE2*	g9736	Chr7:51190093..51190608(+)	0	171	17.75	10.00	0.322	97.02	*PaOLE1*	WGD	46..157	U
*PaOLE3*	g12771	Chr2:32085665..32086144(+)	0	159	17.47	9.74	0.211	98.81	*PaOLE1*	Dispersed	19..126	M
*AcOLE1*	Aqcoe3G048300	Chr3:3052078..3052669(+)	0	167	17.95	9.67	0.257	99.88	-	-	41..153	U
*AcOLE2*	Aqcoe7G144100	Chr7:9197082..9198218(−)	0	150	16.14	9.70	0.112	94.33	*AcOLE1*	Dispersed	24..135	M
*AcOLE3*	Aqcoe3G267500	Chr3:31392522..31393370(+)	0	146	15.38	9.30	0.482	112.81	*AcOLE1*	Dispersed	27..137	SL
*AcOLE4*	Aqcoe7G093500	Chr7:5627997..5628401(−)	0	134	13.90	10.02	0.516	108.43	*AcOLE3*	Dispersed	23..119	SL
*AcOLE5*	Aqcoe3G202700	Chr3:21602227..21603086(−)	0	171	18.12	9.39	0.116	95.85	*AcOLE3*	Dispersed	35..158	SH
*CpOLE1*	sunset09G0006960	Chr9:5118166..5118919(−)	0	167	18.08	9.84	0.396	99.76	-	-	41..152	U
*CpOLE2*	sunset07G0007350	Chr7:6423723..6424251(+)	0	131	13.68	9.56	0.422	108.85	*CpOLE2*	Transposed	17..125	M
*CpOLE3*	sunset09G0008730	Chr9:6575284..6575789(−)	1	136	14.14	9.89	0.347	104.78	*CpOLE1*	Dispersed	15..127	SL
*CpOLE4*	sunset01G0003770	Chr1:3107234..3107629(−)	0	131	13.56	10.89	0.675	122.82	*CpOLE4*	Dispersed	26..129	SH
*CpOLE5*	sunset09G0012790	Chr9:14063375..14064171(+)	0	149	15.96	10.34	0.169	106.71	*CpOLE3*	Dispersed	28..138	SH
*CpOLE6*	sunset04G0023010	Chr4:30227031..30227636(+)	0	145	14.83	5.56	0.720	117.72	*CpOLE3*	Dispersed	27..104	N
*MoOLE1*	-	Scf12:425030..425509(+)	0	159	17.39	10.00	0.348	94.47	-	-	33..145	U
*MoOLE2*	GLEAN_10017149	Scf5:3253661..3254092(+)	0	143	15.25	9.23	0.344	100.35	*MoOLE1*	Dispersed	21..132	M
*MoOLE3*	GLEAN_10002091	Scf132:402521..407622(+)	1	137	14.64	9.89	0.397	112.48	*MoOLE1*	Dispersed	17..127	SL
*MoOLE4*	GLEAN_10017990	Scf4:3104843..3105331(+)	0	162	16.81	10.28	0.355	107.22	*MoOLE4*	γ WGD	37..149	SH
*MoOLE5*	GLEAN_10007003	Scf35:698143..698517(−)	0	124	13.15	9.95	0.784	121.13	*MoOLE6*	Dispersed	29..116	SH
*MoOLE6*	GLEAN_10005491	Scf65:559782..564316(+)	0	267	28.01	6.07	0.471	98.58	*MoOLE3*	Dispersed	25..123 171..265	N
*BsOLE1*	BsiG0022789	Chr4:24719773..24720252(−)	0	159	17.43	9.84	0.394	102.45	-	-	33..145	U
*BsOLE2*	BsiG0031356	Chr5:115702358..115702837(+)	0	159	17.55	9.69	0.424	106.73	*BsOLE1*	Dispersed	33..145	U
*BsOLE3*	BsiG0027711	Chr5:12196723..12197160(+)	0	145	15.60	9.52	0.239	98.97	*BsOLE1*	Dispersed	21..131	M
*BsOLE4*	BsiG0023505	Chr4:38309092..38309529(−)	0	145	15.48	9.55	0.374	106.28	*BsOLE3*	α WGD	21..132	M
*BsOLE5*	BsiG0007300	Chr1:161375811..161376341(−)	1	139	14.61	9.77	0.397	103.17	*BsOLE1*	Dispersed	18..128	SL
*BsOLE6*	BsiG0026540	Chr4:140933592..140934122(−)	1	135	14.12	9.52	0.400	105.48	*BsOLE5*	α WGD	13..124	SL
*BsOLE7*	BsiG0026541	Chr4:140939897..140940427(−)	1	135	14.08	9.52	0.417	107.63	*BsOLE6*	Tandem	13..124	SL
*BsOLE8*	BsiG0004876	Chr1:127499323..127499826(−)	0	167	17.89	9.39	0.180	107.49	*BsOLE5*	Dispersed	38..147	SH
*BsOLE9*	BsiG0006900	Chr1:156712314..156712805(+)	0	163	17.58	9.51	-0.144	90.98	*BsOLE8*	γ WGD	34..138	SH
*BsOLE10*	BsiG0025867	Chr4:130692072..130692554(−)	0	160	17.10	9.97	0.078	104.25	*BsOLE9*	α WGD	23..133	SH
*CsOLE1*	Cs02G002030	Chr2:9103395..9103838(+)	0	147	15.81	9.35	0.563	112.18	-	-	26..133	U
*CsOLE2*	Cs15G003740	Chr15:2440935..2441444(+)	0	146	15.87	9.68	0.205	88.90	*CsOLE1*	Dispersed	21..132	M
*CsOLE3*	Cs12G001310	Chr12:683951..684453(+)	1	115	11.92	11.00	0.738	123.83	*CsOLE1*	Dispersed	16..114	SL
*CsOLE4*	Cs06G005610	Chr6:16853769..16854634(−)	1	149	15.51	9.99	0.170	94.36	*CsOLE3*	β WGD	21..132	SL
*CsOLE5*	Cs02G003290	Chr2:9784653..9785174(+)	1	134	14.18	10.20	0.352	108.51	*CsOLE3*	α WGD	16..128	SL
*CsOLE6*	Cs01G009580	Chr1:23015766..23016309(−)	1	149	15.40	9.59	0.358	110.13	*CsOLE7*	α WGD	34..147	SH
*CsOLE7*	Cs02G004630	Chr2:10712950..10713929(−)	1	161	16.81	9.69	0.103	99.44	*CsOLE3*	Dispersed	32..148	SH
*CsOLE8*	Cs14G006880	Chr14:6118834..6119582(+)	1	152	16.32	9.69	0.245	100.07	*CsOLE7*	β WGD	29..140	SH
*CvOLE1*	Clevi.0032s0439	Scf32:644486..645807(−)	1	159	17.04	9.75	0.383	103.14	-	-	38..145	M
*CvOLE2*	Clevi.0001s1658	Scf1:3646632..3647045(−)	0	137	14.65	9.61	0.412	104.01	*CvOLE1*	Dispersed	19..128	SL
*CvOLE3*	Clevi.0015s0023	Scf15:3700181..3701021(−)	1	143	14.98	10.20	0.262	96.22	*CvOLE1*	Dispersed	19..130	SL
*CvOLE4*	Clevi.0004s1912	Scf4:2316861..2318229(+)	1	157	16.42	9.98	0.297	104.33	*CvOLE6*	Dispersed	32..144	SH
*CvOLE5*	Clevi.0042s0814	Scf42:1538196..1539122(−)	1	161	16.79	9.69	0.441	106.02	*CvOLE4*	γ WGD	35..147	SH
*CvOLE6*	Clevi.0015s0551	Scf15:942332..943312(−)	0	145	14.89	5.25	0.560	106.34	*CvOLE3*	Dispersed	27..132	N
*GgOLE1*	GG13G018590	Chr13:9683189 9684092(+)	1	164	17.30	9.57	0.410	104.15	-	-	43..150	U
*GgOLE2*	GG05G000440	Chr5:273726 274730(−)	1	162	17.16	9.41	0.446	107.72	*GgOLE1*	Dispersed	41..148	U
*GgOLE3*	GG07G021290	Chr7:10989271 10989687(+)	0	138	14.79	9.72	0.442	110.22	*GgOLE1*	Dispersed	19..128	M
*GgOLE4*	GG02G144870	Chr2:66911747 66912474(+)	1	144	15.05	10.20	0.272	96.94	*GgOLE1*	Dispersed	19..130	SL
*GgOLE5*	GG05G049880	Chr5:23864915 23865662(+)	1	159	16.78	9.89	0.302	104.91	*GgOLE7*	Dispersed	34..146	SH
*GgOLE6*	GG15G098790	Chr15:45889738 45890318(−)	1	161	16.77	9.52	0.406	109.01	*GgOLE5*	Dispersed	35..147	SH
*GgOLE7*	-	Chr6:796424..801985(−)	1	139	14.63	4.43	0.609	110.79	*GgOLE4*	Dispersed	26..124	N
*ThOLE1*	LOC104821850	Scf34:1318549..1319624(−)	1	155	16.50	9.63	0.665	113.81	-	-	34..141	U
*ThOLE2*	LOC104819676	Scf3:6116766..6117891(−)	1	156	16.49	9.39	0.485	105.64	*ThOLE1*	Dispersed	35..142	U
*ThOLE3*	LOC104818593	Scf3:633964..634782(−)	0	138	14.70	9.56	0.449	106.81	*ThOLE1*	Dispersed	19..128	M
*ThOLE4*	LOC104825056	Scf42:463230..464045(+)	1	144	15.05	10.20	0.332	102.99	*ThOLE1*	Dispersed	19..130	SL
*ThOLE5*	LOC104811538	Scf2:1261264..1262172(−)	1	144	15.22	9.90	0.273	100.28	*ThOLE4*	α WGD	22..133	SL
*ThOLE6*	LOC104805374	Scf11:1401936..1403042(+)	1	159	16.89	9.89	0.177	98.74	*ThOLE8*	Dispersed	34..146	SH
*ThOLE7*	LOC104802395	Scf8:1757388..1758247(+)	1	161	16.80	9.69	0.455	110.81	*ThOLE6*	β WGD	35..147	SH
*ThOLE8*	LOC104811693	Scf2:1907125..1907884(+)	0	142	14.50	6.56	0.492	104.44	*ThOLE4*	Dispersed	28..99	N

**Table 2 plants-13-00280-t002:** Evolutionary rate of WGD repeats identified in this study. Ks and Ka were calculated using PAML. (At: *A. thaliana*; Bs: *B. sinensis*; Cs: *C. spinosa*; Cv: *C. violacea*; Ka: nonsynonymous substitution rate; Ks: synonymous substitution rate; Me: *M. esculenta*; Mo: *M. oleifera*; OLE: oleosin; Pa: *P. americana*; Pt: *P. trichocarpa*; Th: *T. hassleriana*).

Gene1	Gene2	Identity (%)	Ks	Ka/Ks
*PaOLE1*	*PaOLE2*	77.8	0.8501	0.1372
*MoOLE4*	*MoOLE5*	57.7	2.0410	0.1831
*BsOLE3*	*BsOLE4*	89.3	0.2700	0.2629
*BsOLE5*	*BsOLE6*	89.0	0.1962	0.2749
*BsOLE8*	*BsOLE9*	55.0	1.5864	0.9415
*BsOLE9*	*BsOLE10*	76.6	0.2810	0.2839
*CsOLE3*	*CsOLE4*	54.9	1.6273	0.1248
*CsOLE3*	*CsOLE5*	76.5	0.3691	0.1054
*CsOLE6*	*CsOLE7*	75.8	0.5869	0.1216
*CsOLE7*	*CsOLE8*	60.3	-	-
*CvOLE4*	*CvOLE5*	64.0	1.9437	0.1515
*ThOLE4*	*ThOLE5*	86.0	0.3409	0.1610
*ThOLE6*	*ThOLE7*	65.6	1.5677	0.1797
*At*-*Sm1*	*At*-*Sm2*	66.9	1.3093	0.1880
*At*-*S3*	*At*-*S5*	58.2	1.3683	0.1592
*At*-*S1*	*At*-*S4*	53.1	-	-
*At*-*S2*	*At*-*S4*	61.8	1.5782	0.1625
*PtOLE2a*	*PtOLE2b*	75.1	0.3138	0.5186
*PtOLE3a*	*PtOLE3b*	86.2	0.1619	0.8850
*PtOLE4a*	*PtOLE4b*	90.7	0.2091	0.3161
*PtOLE5a*	*PtOLE5b*	84.0	0.3696	0.2675
*MeOLE1a*	*MeOLE1b*	81.1	0.7428	0.1198
*MeOLE3a*	*MeOLE3b*	76.7	0.6126	0.2047
*MeOLE4a*	*MeOLE4b*	78.1	0.4175	0.3827
*MeOLE4b*	*MeOLE5*	59.6	1.7862	0.1548

## Data Availability

SRA accession numbers of transcriptome data used in this study are shown in Appendix A.

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
