# Peer review of "Integrative Analysis of Oleosin Genes Provides Insights into Lineage-Specific Family Evolution in Brassicales"

_plants, 2024, doi:10.3390/plants13020280_

Round 1
Reviewer 1 Report
Comments and Suggestions for Authors
Significant information seems to be missing on how new oleosin genes were identified? Is the homology among them so high that they can be found by blast search? Were all of these genes already identified? If so, why was this publication put together? What genes were identified in this study and what genes were previously found? It is difficult to evaluate this publication when I can't figure out why the authors did and what results preceded theirs. This issue needs to be clarified before I can evaluate the merit of this submission.
The readers shouldn't need to understand the relationships among all of the plant families to figure out the importance of the findings. This is not a taxonomy journal. I know the authors covered some of the important evolutionary relationships, but more reminders on the evolutionary position of genera and species are needed.
Comments on the Quality of English LanguageFine
Author Response
Response to Reviewer #1:
Comment: Significant information seems to be missing on how new oleosin genes were identified? Is the homology among them so high that they can be found by blast search? Were all of these genes already identified? If so, why was this publication put together? What genes were identified in this study and what genes were previously found? It is difficult to evaluate this publication when I can't figure out why the authors did and what results preceded theirs. This issue needs to be clarified before I can evaluate the merit of this submission.
Response: As described in the part of “4.1. Sequence retrieval and identification of oleosin family genes”, in this study, oleosin homologs were identified using HMMER (v3.3, http://hmmer.janelia.org/) rather than blast search for its high sensitivity. Indeed, compared with previous studies, more oleosin homologs were identified in this study, e.g., six and eight vs five and seven in papaya and spider flower, respectively. Further MOTIF Search confirmed the presence of the conserved oleosin domain in deduced peptides of these new genes, supporting that they are true oleosin homologs. Actually, all oleosin genes analyzed in this study were identified using HMMER with updated genomes, including arabidopsis, poplar, and cassava. Thereby, they may also be considered to independently identified oleosin homologs, though they were supplemented in Table S1. Oleosin genes identified in arabidopsis, poplar, and cassava were included for comparison mainly for their taxonomy positions and chromosome-level genome assemblies.
Comment: The readers shouldn't need to understand the relationships among all of the plant families to figure out the importance of the findings. This is not a taxonomy journal. I know the authors covered some of the important evolutionary relationships, but more reminders on the evolutionary position of genera and species are needed.
Response: As shown below, in the revised version of manuscript, an updated Figure 2 was employed to display the taxonomy relationships of tested species, which also shows species-specific distribution of nine orthogroups (i.e., U1, U2, SL, SH1, SH2, SH3, M, N, and T) identified in this study.

Reviewer 2 Report
Comments and Suggestions for Authors
Authors presented an integrative analysis of oleosin genes from five families of Brassicales, an economically important order of flowering plants. Oleosins are a class of highly abundant structural proteins of lipid droplets (LDs), 40 which represent a major carbon reserve and are widely present in various plant organs 41 such as seeds, pollen, flowers, fruits, and certain tubers. Oleosin genes have diverged into at least six clades known (P, U, SL, SH, T, and M) during later evolution. The origin and presence of each of these group in the Brassicales order is studied, making a focus in the Oleosin genes coded by the papaya genome (Carica papaya, Caricaceae). This an interesting and well described study that include several approaches (phylogetic analysis, syntheny analysis, gene structure analysis and expression analysis) to classify several Oleosin gene belonging to the Brassicales order.
Minor remarks:
- In the introduction, one sentence should be included to explain more in detail the special attention to the papaya Oleosin genes. It is mentioned that it is a non-model plant but, there are other raison to be interested on this king of genes in Carica papaya?
- Can authors explain why the phylogenetic three in Fig.1 is unrooted? In general, a rooted tree attempts to identify when various species diverged from a common ancestor, while the unrooted tree does not.
- This is also in relation with the new identified N clade, which is described as an early originated. In that sense we can wondering if it is close to the primitive (P) clade (present only in green algae, mosses, and ferns) or to the others ones?
- Figure 3 it is a bit complicated to follow. Could the authors give a specific color to each specie, instead of change it for each plot?
- It is not clear in the material and methods if a RNA-seq was done for Carica papaya or only RT-PCR? Please provide more details about the RNA extraction, tissues, and all procedure for data analysis of RNA-seq data.
- Sentence on line 205-209 is complicated to understand, please rephase.
- Some abbreviations are not included on the good order
-PaOLE genes in line 99
-OGs in line 166
- Please homogenize the name of Arabidopsis (first letter in capitals)
Author Response
Response to Reviewer #2:
Comment: Authors presented an integrative analysis of oleosin genes from five families of Brassicales, an economically important order of flowering plants. Oleosins are a class of highly abundant structural proteins of lipid droplets (LDs), which represent a major carbon reserve and are widely present in various plant organs such as seeds, pollen, flowers, fruits, and certain tubers. Oleosin genes have diverged into at least six clades known (P, U, SL, SH, T, and M) during later evolution. The origin and presence of each of these group in the Brassicales order is studied, making a focus in the Oleosin genes coded by the papaya genome (Carica papaya, Caricaceae). This an interesting and well described study that include several approaches (phylogetic analysis, syntheny analysis, gene structure analysis and expression analysis) to classify several Oleosin gene belonging to the Brassicales order.
Minor remarks:
- In the introduction, one sentence should be included to explain more in detail the special attention to the papaya Oleosin genes. It is mentioned that it is a non-model plant but, there are other raison to be interested on this king of genes in Carica papaya?
Response: As suggested by the reviewer, more information about papaya has been supplemented as “In papaya, an economically and nutritionally important tree fruit crop widely cultivated in tropical and subtropical areas [18], this group was shown to be constitutively expressed”
Comment: Can authors explain why the phylogenetic three in Fig.1 is unrooted? In general, a rooted tree attempts to identify when various species diverged from a common ancestor, while the unrooted tree does not.
Response: Evolutionary tree presented in Figure 1 was employed to clarify evolutionary relationships of oleosin genes in A. trichopoda, P. americana, A. coerulea, C. papaya, M. oleifera, B. sinensis, C. spinosa, C. violacea, G. gynandra, T. hassleriana, M. esculenta, P. trichocarpa, and A. thaliana. In fact, it is not suitable to construct a rooted tree since oleosin genes have already diverged into five clades in A. trichopoda.
Comment: This is also in relation with the new identified N clade, which is described as an early originated. In that sense we can wondering if it is close to the primitive (P) clade (present only in green algae, mosses, and ferns) or to the others ones?
Response: Compared with Clade N, Clade U, which is typical for the C-terminal AAPGA and universally present in all land plants including Selaginella moellendorffii, is more closer to Clade P. Clade N is more likely to be derived from Clade SL, appeared sometime before the radiation of angiosperm, though the exact time needs to be further studied.
Comment: Figure 3 it is a bit complicated to follow. Could the authors give a specific color to each specie, instead of change it for each plot?
Response: This figure presents both intra- and inter-synteny results. Since more than five species were included, it will be hard to distinguish when a color is assigned to each species. Thereby, only two colors, i.e., red (intra) and purple (inter), were assigned to syntenic blocks that contain oleosin genes.
Comment: It is not clear in the material and methods if a RNA-seq was done for Carica papaya or only RT-PCR? Please provide more details about the RNA extraction, tissues, and all procedure for data analysis of RNA-seq data.
Response: In this study, tissue-specific expression profiles of CpOLE genes were investigated on the basis of RNA-seq data available in NCBI. By contrast, qRT-PCR was used to analyze gene expression profiles in the seed tissue, where oleosin genes are usually abundant.
Comment: Sentence on line 205-209 is complicated to understand, please rephase.
Response: It has been revised.
Comment: Some abbreviations are not included on the good order
-PaOLE genes in line 99
-OGs in line 166
Response: It has been revised.
Comment: Please homogenize the name of Arabidopsis (first letter in capitals)
Response: It has been revised.

Reviewer 3 Report
Comments and Suggestions for Authors
Journal: Plants (ISSN 2223-7747)
Manuscript ID: plants-2726246
Title: Integrative analysis of oleosin genes provides insights into lineage-specific family evolution in Brassicales
I have thoroughly reviewed the submitted manuscript that focuses on the evolutionary study of oleosins (OLEs) in Brassicales. The manuscript presents a detailed analysis of the oleosin genes across various families within the order Brassicales, contributing valuable information to the field of plant molecular biology and genetics. However, few aspects need to be addressed to strengthen the manuscript:
The introduction and discussion sections would benefit from a more extensive review of the literature, especially recent studies that could provide context and relevance to the current findings.
The methods employed for evolutionary and homologous analyses are briefly mentioned, but there is a lack of depth and clarity. The authors should provide more detailed information about the techniques and parameters used in these analyses to enable reproducibility and better understanding.
The manuscript is based on bioinformatic analyses, if possible try to perform some form of experimental validation, such as gene expression analysis in non-model plants, which may add significant value to the study.
The manuscript discusses the classification of oleosin genes into clades and orthogroups, the phylogenetic analysis appears to be superficial. I suggest need of comprehensive phylogenetic analysis, including tree construction and branch support values, that may enhance the validity of the clade classification.
The manuscript presents findings on the lineage-specific expansion and expression patterns of oleosin genes in Brassicales. However, the functional implications of these findings are not adequately discussed. The authors needs to elaborate on how these evolutionary patterns might affect the biological functions of oleosins in different plant tissues or developmental stages.
The study lacks a comparative analysis with other orders or families outside Brassicales. If possible make comparative study that could provide a broader perspective on the evolution and diversification of oleosin genes in angiosperms.
Some of the figures, particularly those representing gene expression patterns, are not very clear and could be improved for better visualization and interpretation.
Comments on the Quality of English Language
English quality is good
Author Response
Response to Reviewer #3:
Comment: Manuscript ID: plants-2726246
Title: Integrative analysis of oleosin genes provides insights into lineage-specific family evolution in Brassicales
I have thoroughly reviewed the submitted manuscript that focuses on the evolutionary study of oleosins (OLEs) in Brassicales. The manuscript presents a detailed analysis of the oleosin genes across various families within the order Brassicales, contributing valuable information to the field of plant molecular biology and genetics. However, few aspects need to be addressed to strengthen the manuscript:
Comment: The introduction and discussion sections would benefit from a more extensive review of the literature, especially recent studies that could provide context and relevance to the current findings.
Response: Thank the reviewer for the positive consideration for our manuscript. We have tried to update recent literatures.
Comment: The methods employed for evolutionary and homologous analyses are briefly mentioned, but there is a lack of depth and clarity. The authors should provide more detailed information about the techniques and parameters used in these analyses to enable reproducibility and better understanding.
Response: It has been revised.
Comment: The manuscript is based on bioinformatic analyses, if possible try to perform some form of experimental validation, such as gene expression analysis in non-model plants, which may add significant value to the study.
Response: In this study, a focus on comparative analysis of the oleosin gene family in Brassicales was conducted, which includes 13 species representing ten plant families. Actually, gene expression analysis was conducted via both transcriptome profiling and qRT-PCR.
Comment: The manuscript discusses the classification of oleosin genes into clades and orthogroups, the phylogenetic analysis appears to be superficial. I suggest need of comprehensive phylogenetic analysis, including tree construction and branch support values, that may enhance the validity of the clade classification.
Response: It is well known that phylogenetic analysis of highly divergent gene families such as oleosin is challenging. In this study, classification of oleosin genes into clades and orthogroups is not only supported by evolutionary analysis, but also by BRH-based orthologous and synteny analyses.
Comment: The manuscript presents findings on the lineage-specific expansion and expression patterns of oleosin genes in Brassicales. However, the functional implications of these findings are not adequately discussed. The authors needs to elaborate on how these evolutionary patterns might affect the biological functions of oleosins in different plant tissues or developmental stages.
Response: As suggested by the reviewer, functional implications have been supplemented as “In Brassicales, lineage-specific expansion and tissue-specific expression of oleosin genes reflect their roles in oil accumulation of seeds and anther [3,34]. In seeds, accumulation of oleosins are usually negatively correlated with LD size but positively associated with oil content, which could not only affect seed germination but also the freezing tolerance of seeds [34,35]. Moreover, Brassicaceae-specific T oleosins are acquired for tapetosome formation and conferred additive benefit of pollen vigor [44].”.
Comment: The study lacks a comparative analysis with other orders or families outside Brassicales. If possible make comparative study that could provide a broader perspective on the evolution and diversification of oleosin genes in angiosperms.
Response: Thank the reviewer for the helpful suggestion. This work is ongoing. We hope it will come to the readers in the near future.
Comment: Some of the figures, particularly those representing gene expression patterns, are not very clear and could be improved for better visualization and interpretation.
Response: It has been revised.

Round 2
Reviewer 1 Report
Comments and Suggestions for Authors
The authors addressed both of my concerns. I was interested why HMMER performed better than other homology searches. The first cited reference didn't explain this. The link for the second reference was not functional. I suggest updating the text with an explanation and the reference with the correct DOI link.
Comments on the Quality of English LanguageNone
Author Response
Response to Reviewer #1:
Comment: The authors addressed both of my concerns. I was interested why HMMER performed better than other homology searches. The first cited reference didn't explain this. The link for the second reference was not functional. I suggest updating the text with an explanation and the reference with the correct DOI link.
Response: There are two main methods for identification of protein homologs, i.e. BLASTP and HMMER. Whereas BLASTP is based on homologous alignment of whole protein sequences, HMMER is more popular and sensitive for homologous alignment of conserved domains, e.g., the oleosin domain profile (PF01277) as used in this study.
We have checked DOI links of all references cited in this study, and found that they are all functional. Moreover, we have updated the link of reference 59, one paper published in Chinese.